# OG-SNR : Open-set graph learning with structural noise robustness

## Abstract

Open-set graph learning aims to training graph-based models that can accurately classify known in-distribution classes while identifying and handing previously unknow classes during inference. It is critical for high-stakes, real-world applications where models frequently encounter unexpected data, including finance, security and healthcare etc. Though utilizing the message-passing mechanism, Graph Neural Networks (GNNs) have demonstrated outstanding performance in this area by focusing on preserving structural properties, the robustness against noises is generally ignored, especially the structural noise which is inevitable in open-world scenarios of real-world applications. In this paper, we propose a novel framework to achieve open-set node classification with structure-noise robustness, which is specifically tailored for open-set graph data with structural noise and out-of-distribution (OOD) classes. Specifically, our approach refines graph structures by leveraging both node features and structural information. Furthermore, we mitigate the impact of noisy edges using a curriculum learning framework and dynamically select a subset of samples as "pseudo-OOD nodes" during training. By incorporating an entropy maximization loss, the method achieves open-set node classification guided by node confidence scores. To the best of our knowledge, this is the first study to explore this problem. Extensive experiments validate the superiority of our proposed approach.

## 1 Introduction

Node classification involves assigning labels to unlabeled nodes in a graph. This task is crucial in numerous applications, including predicting traffic states (Zheng et al., 2020), identifying diseases (Chereda et al., 2019), and completing user profiles on social networks (Wong et al., 2021). Methods based on GNNs (Zhu et al., 2022a; Liu et al., 2021b; Zhang et al., 2018) have achieved outstanding performance in tackling this problem, with a common assumption in the design of these methods is the availability of a complete label set ( *known classes*) during the training phase.

However, **this closed-set assumption often does not hold in real-world scenarios** (Zhang et al., 2022a; Wu et al., 2024; Zhang et al., 2024a). For instance, a Graph Neural Network (GNN) trained on existing data and applied to data from scientific social networks may encounter researchers contributing to new conferences that were not observed during training. When faced with samples from unknown classes, traditional GNN methods (Hamilton et al., 2017; Kipf & Welling, 2016a; Veličković et al., 2017) may incorrectly assign labels from known classes to these nodes, which significantly impacts the classification performance. Therefore, it is necessary to develop models that can accurately classify known classes while effectively rejecting unknown classes. This task is referred to as the open-set node classification problem (Zhang et al., 2022b). Additionally, **GNNs are highly sensitive to the quality of the given graph structures**. Noisy or incomplete graphs often lead to unsatisfactory representations and prevent us from fully understanding the mechanism underlying the system.

Moreover, **open-world scenarios often involves changes in graph structures**. Recent studies (Zügner et al., 2018; Zügner & Günnemann, 2019b) find that GNNs are particularly vulnerable to adversarial attacks on graph data. Specifically, adversarial attacks on graphs usually inject small, subtle changes into the graph structure and node features, which can easily deceive GNNs into making incorrect predictions. Even minor changes in the graph, such as adding several edges (Zügner

& Günnemann, 2019b), can obviously degrade the predictive performance of GNN models. This presents a significant challenge for the use of GNNs in practical applications, especially in high-risk scenarios such as medical analysis. Hence, several methods have been proposed to enhance the generalization ability of models trained with structural noise, collectively referred to as graph structure learning (Li et al., 2022b; Zhu et al., 2024; In et al., 2024).

Nevertheless, as far as we know, there is not work explore the problem of structure-noise-robust open-set node classification, where both subtle changes of graph data, such as adversarial attacks, and harsh changes, such as out-of-distribution (OOD) nodes, exist simultaneously. Therefore, in this work, we try to develop a robust open-set node classification method, named OG-SNR, that capable of both open-set recognition and mitigating the effects of structural noise. To achieve this, an encoder capable of effectively learning both node feature and structural information is employed to encode node embeddings and use them to weight edge connections. Based on the homogeneity principle of graphs, the graph structure is refined. Furthermore, we introduce a curriculum learning paradigm, quantifying all edges in the graph and progressively expanding the graph structure for learning according to the difficulty of the edges. During training, pseudo-unknown nodes are generated, and the output of the model is balanced using a combination of label and entropy maximization losses. Finally, the threshold for open-set recognition is automatically calculated based on the output scores of known-class and pseudo-unknown-class samples. To the best of our knowledge, OG-SNR represents an early attempt at open-set node classification robust to structural noise entropy. Extensive experiments on benchmark graph datasets demonstrate the superiority of OG-SNR.

## 2 PROBLEM DEFINITION & PRELIMINARY

This paper investigates the open-set node classification problem within a graph. We represent the graph as $G = (V, E, X)$, where $V$ denotes a set of $N$ nodes, specifically $V = \{v_i | i = 1, \ldots, N\}$. $E$ represents the set of edges connecting node pairs, defined as $E = \{e_{i,j} | i, j = 1, \ldots, N, i \neq j\}$. Each node's features are captured in the feature matrix $X \in \mathbb{R}^{N \times d}$, with $d$ representing the feature dimension. The graph's topology is encoded by the adjacency matrix $A \in \mathbb{R}^{N \times N}$, where $A_{i,j} = 1$ indicates an edge between nodes $v_i$ and $v_j$, and $A_{i,j} = 0$ otherwise. Node labels are represented by the matrix $Y \in \mathbb{R}^{N \times C}$, where $C$ signifies the number of known node classes. If node $v_i$ belongs to class $c$, then $y_{i,c} = 1$, and $y_{i,c} = 0$ otherwise.

In the **open-set node classification problem**, a GNN encoder, denoted as $\phi_a$, processes node features $X$ and the adjacency matrix $A$. This encoder aggregates neighborhood information to generate node representations. And given a graph $G = (V, E, X)$, $\mathcal{D}_{tr} = (X_{tr}, Y_{tr})$ represents the training node data. The test nodes are denoted by $\mathcal{D}_{te} = (X_{te}, Y_{te})$, where $X_{te} = S \cup U$ and $Y_{te} = \{1, \ldots, C, C+1, \ldots\}$. The set $S$ is the nodes that belong to known classes that already appeared in $\mathcal{D}_{tr}$ and $U$ is the set of nodes that do not belong to any known class (i.e., unknown class nodes). The objective of open-set node categorization is to develop a $(C+1)$-class classifier $\phi_b$ such that $f(\phi_a, \phi_b; X_{te}, A) : \{X_{te}, A\} \to \overline{\mathcal{Y}}$, by minimizing the expected risk $f^* = arg \min_{f \in \mathcal{H}} \mathbb{E}_{(x,y) \sim \mathcal{D}_{te}} \mathbb{I}(y \neq f(\phi_a, \phi_b; x, A))$, where $\overline{\mathcal{Y}} = \{1, \ldots, C, unknown\}$ (Yu et al., 2017).

When exploring approaches for open-set classification, one intuitive strategy (Hendrycks & Gimpel, 2016) for addressing open-set classification involves leveraging confidence thresholds for decision-making. Typically, the maximum predicted probability of a sample across all closed-set classes is used as the confidence score. Mathematically, the confidence score $conf_i$ for a sample $x_i$ is expressed as $\max_{c=1,\ldots,C} f_c(x_i, A)$, where $f_c(x_i, A)$ represents the output probability of the $c$-th class for the sample $x_i$ under the graph structure $A$, and $C$ is the total number of closed-set classes. This approach assumes that the model's predictions are confident for closed-set instances but less reliable for open-set ones. Based on this confidence score, an open-set classifier can be defined as:

$$\hat{y}_i = \begin{cases} \arg\max_{c=1,\ldots,C} f_c(x_i, A), & \text{if } conf_i > \tau, \\ \text{unknown}, & \text{otherwise.} \end{cases} \quad (1)$$

Here, $\tau$ denotes a predefined confidence threshold. If the maximum confidence score of a sample exceeds $\tau$, it is assigned to the class with the highest predicted probability; otherwise, it is designated as "unknown". However, the overconfidence inherent in deep neural networks (Bendale & Boult, 2016), particularly for misclassified or open-set samples, often renders this thresholding

approach unreliable. Confidence scores are frequently inflated for both in-distribution and out-of-distribution samples, resulting in significant overlap and insufficient discrimination between known and unknown categories.

In the **open-set node classification with structure noise** problem, Given a graph $G = (V, A, X)$, real-world graphs are often susceptible to various types of nosie. These nosie inevitably alter the graph structure, denoted as $\tilde{A}$, which often leads to a significant decline in the performance of GNNs trained on such perturbed graphs. The objective of open-set node classification on graphs with structural noise is to learn a robust GNN-based node classification model using $X$ and $\tilde{A}$ that not only achieves high classification accuracy on in-distribution (IND) data but also effectively identifies out-of-distribution nodes by minimizing the expected risk $\overline{f}^* = arg\min_{f \in \mathcal{H}} \mathbb{E}_{(x,y) \sim \overline{\mathcal{D}}_{te}} \mathbb{I}(y \neq f(\phi_a, \phi_b; x, \tilde{A}))$. Specifically, in the training set, all samples are IND samples, and their true labels are from the known classes (belonging to $S$). However, the test set $\mathcal{D}_{te}$ primarily comprises IND nodes but also includes OOD node(with the true class being $C + 1$). Thus, the goal is to train a robust open-set node classification model that assigns IND nodes to their respective classes, $\bar{y}_i \in \{1, \ldots, C\}$, where $\bar{y}_i$ is the predicted label for node $i$. The model should also classify OOD nodes into an "unknown" category, $\bar{y}_i = C + 1$.

## 3 METHODOLOGY

To address the challenge of open-set node classification in graphs with structural noise, we propose a novel framework, **OG-SNR**, which is designed to learn an effective open-set node classifier from noisy graph data. As illustrated in Figure 1, the framework operates in two main stages. In the *pre-training stage*, node representations generated by an encoder, which integrates both feature and structural information, are employed to refine the graph structure under the principle of homophily, thereby producing a cleaner graph. In the *fine-tuning stage*, a curriculum learning mechanism prioritizes the learning of easily distinguishable edges, which enhances the model's robustness. During training, a subset of high-entropy known nodes is treated as pseudo-unknown nodes to compute an *entropy maximization loss*, ensuring a balanced confidence distribution in the model's predictions. During testing, the framework dynamically determines a reliable threshold by combining the confidence scores of known samples, high-entropy known samples, and pseudo-unknown samples. This adaptive threshold enables effective open-set node classification. The proposed framework integrates these components and jointly optimizes them using *label loss* and *entropy maximization loss*, thereby improving both robustness and accuracy in open-set scenarios.

### 3.1 PRE-TRAINING FOR GRAPH STRUCTURE REFINEMENT

As demonstrated in IG-JSMA (Wu et al., 2019) and NIPA (Zügner & Günnemann, 2019a), the most detrimental graph structure perturbations for GNN performance arise from the presence of edges connecting nodes with different labels and the absence of edges between nodes with the same label. Due to the message-passing mechanism, these perturbations can significantly impair the ability of GNNs to learn effective representations. To address these issues, we adopt an encoder designed to efficiently learn high-quality representations and refine the graph structure, thereby mitigating these pitfalls. The refinement process reduces heterophilous edges (edges connecting nodes of different classes) while adding appropriate homophilous edges (edges connecting nodes within the same class), under the homophily assumption that similar nodes are more likely to belong to the same class (Abu-El-Haija et al., 2019; Chien et al., 2020; Yifan et al., 2020). Node similarity is thus leveraged to identify potential homophilous and heterophilous edges.

Specifically, we employ a two-layer Graph Convolutional Network (GCN) as the node feature encoder $f_\theta$. This encoder integrates both node attributes $X$ and structural information from the adjacency matrix $A$, thereby aggregating neighborhood features to produce representative node embeddings $\mathbf{H}$. Formally, the process can be expressed as:

$$\mathbf{H} = f_\theta(X, A), \tag{2}$$

After obtaining high-quality node representations, the next step is to refine the graph structure. The objective is to improve graph homogeneity by removing heterophilous edges and reinforcing homophilous ones. First, we compute the similarity matrix $M_{ij} = \text{sim}(\mathbf{h}_i, \mathbf{h}_j)$ based on the learned

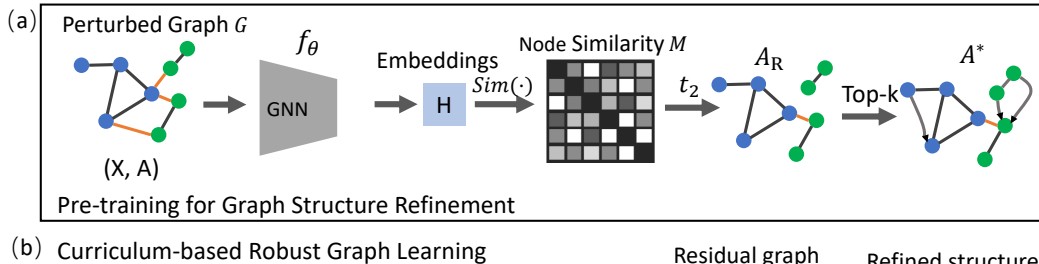

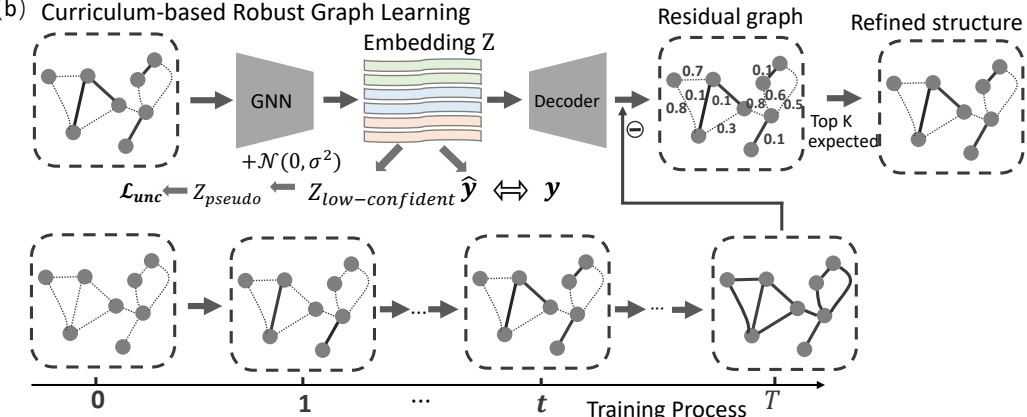

Figure 1: The overall framework of OG-SNR consists of two modules: (a) The Pre-training for Graph Structure Refinement module extracts node embeddings through a GNN encoder and refines the graph structure based on node similarity. (b) The Curriculum-based Robust Graph Learning module extracts latent embeddings using the GNN model, generates pseudo-OOD nodes with embeddings, and applies entropy maximization loss for open-set graph learning. It then reconstructs the input structure using the embeddings and a decoder. A small residual error on an edge indicates a well-predicted dependency, which can be incorporated into the refined structure for the next iteration.

node representations, where $\mathbf{h}_i$ and $\mathbf{h}_j$ are the embeddings of nodes $v_i$ and $v_j$. To mitigate the impact of heterophilous edges, we prune edges with similarity below a threshold $\epsilon$, yielding the pruned adjacency matrix $\mathbf{A}_{pruned}$:

$$\mathbf{A}_{pruned}^{ij} = \begin{cases} 1, & \text{if } M_{ij} > \epsilon \text{ and } A_{ij} = 1, \\ 0, & \text{otherwise.} \end{cases} \tag{3}$$

This step removes edges connecting nodes with dissimilar representations, thereby improving graph homogeneity.

To further enhance the structural quality of the graph, we introduce an edge addition strategy. For each node, we identify its top-$k$ most similar nodes and add edges connecting them. The adjacency matrix for the added edges is defined as:

$$\mathbf{T}_k^{ij} = \begin{cases} 1, & \text{if } v_j \text{ is among the top-}k \text{ similar nodes to } v_i, \\ 0, & \text{otherwise.} \end{cases} \tag{4}$$

Note that $\mathbf{T}_k$ is not symmetric because node similarity is not necessarily reciprocal. For example, $v_i$ may be the most similar node to $v_j$, but not vice versa. Since pruning alone cannot eliminate all perturbations, these added edges help mitigate the residual harmful connections. Empirically, this strategy is particularly effective when the perturbation rate is high.

Finally, the refined adjacency matrix is obtained as $\mathbf{A}^* = \mathbf{A}_{pruned} + \mathbf{T}_k$. This refinement preserves the essential structural properties of the original graph while reinforcing connections between highly similar nodes, thereby improving the effectiveness of message passing in GNNs.

## 3.2 FINE-TUNING VIA CURRICULUM-BASED FOR ROBUST GRAPH LEARNING

Through graph structure refinement, most structural noise can be significantly reduced; however, some noise inevitably remains. To further mitigate the effect of residual noise, we propose a

**curriculum-based edge learning strategy** to enhance robustness. Inspired by human learning patterns, curriculum learning prioritizes simple samples at the early stage and progressively incorporates more complex ones. Since hard-to-learn edges are more likely to correspond to structural noise, this strategy improves robustness by thoroughly fitting "simple and clean" edges while limiting the influence of "complex and noisy" ones.

As shown in Figure 1, at iteration $t$, the current adjacency matrix is fed into the encoder to obtain latent node embeddings, which are then used by a decoder to reconstruct the graph structure via residual fitting. The residual graph $\mathbf{R}$ represents the discrepancy between the refined adjacency matrix $\mathbf{A}^*$ and the reconstructed graph $\hat{\mathbf{A}}^{(t)}$. A smaller residual error indicates a higher probability that the edge aligns with the model's expectation.

Formally, at each iteration $t$, we input the adjacency matrix $\mathbf{A}^{(t)}$ into the GNN encoder $f^{(t)}$ to obtain embeddings:

$$\mathbf{Z}^{(t)} = f^{(t)}(\mathbf{A}^{(t)}, \mathbf{X}), \tag{5}$$

where $\mathbf{X} \in \mathbb{R}^{N \times d}$ is the node feature matrix, $\mathbf{Z}^{(t)} \in \mathbb{R}^{N \times d'}$ denotes the learned node embeddings, and $d'$ is the embedding dimension.

To control edge selection, we introduce a learnable binary mask matrix $\mathbf{S}$, with each element $S_{ij} \in \{0, 1\}$. The training adjacency matrix is defined as $\mathbf{A}^{(t)} = \mathbf{S}^{(t)} \odot \mathbf{A}^*$. To minimize noisy edges, the learning objective is given by:

$$\min_{\mathbf{w}} \mathcal{L}_{\text{osc}} + \beta \sum_{i,j} S_{ij} R_{ij}, \quad \text{s.t.} \; \|\mathbf{S}\|_1 \geq K, \tag{6}$$

where $\mathcal{L}_{\text{osc}}$ is the task-specific loss (e.g., node classification loss in Equation equation 8), the second term penalizes residual errors of selected edges, $K$ constrains the number of retained edges, and $\beta$ balances the two losses.

Since $\mathbf{S}$ is binary, direct optimization is intractable. We therefore relax $\mathbf{S}$ into a continuous domain $[0, 1]$. Noting that $\|\mathbf{S}\|_1 \geq K$ is equivalent to $\|\mathbf{S}\|_1 = K$ due to the non-negativity of $\mathbf{R}$ and $\mathbf{S}$, the objective can be reformulated using a Lagrangian relaxation $\mathcal{L} = \mathcal{L}_{\text{osc}} + \beta \sum_{i,j} S_{ij} R_{ij} - \lambda(\|\mathbf{S}\|_1 - K)$. To simplify optimization, we replace the constraint with a regularization term $g(\mathbf{S}; \lambda)$, yielding:

$$\min_{\mathbf{w}, \mathbf{S}} \mathcal{L}_{\text{osc}} + \beta \sum_{i,j} S_{ij} R_{ij} + g(\mathbf{S}; \lambda), \tag{7}$$

where $g(\mathbf{S}; \lambda) = \lambda \|\mathbf{S} - \mathbf{A}^*\|$. As $\lambda \to \infty$, the mask matrix gradually converges to $\mathbf{A}^*$, automatically incorporating more edges into the training adjacency matrix. This mechanism ensures that only the top-$K$ well-expected edges are retained, while progressively adding edges during training to improve robustness.

### 3.3 UNCERTAINTY LEARNING VIA ENTROPY MAXIMIZATION FOR OPEN-SET CLASSIFICATION

To enable the classifier to distinguish OOD samples, we introduce an entropy maximization loss in addition to the label loss. Under the joint constraints of the two losses, the classifier is encouraged to produce distinct confidence scores for samples from known and unknown classes. The label loss quantifies the divergence between the predicted and ground-truth distributions, thereby guiding the model toward accurate classification. However, because samples from unknown classes only appear in the test set, the label loss is ineffective for these cases. To address this limitation, we designate low-confidence and high-entropy samples from the known classes as pseudo-unknown samples.

Unlike the label loss, the entropy maximization loss balances the classification outputs for each sample, yielding superior performance for unknown samples and improving the model's generalization ability. Specifically, the overall objective of open-set classification, denoted as $\mathcal{L}_{\text{osc}}$, is formulated as:

$$\mathcal{L}_{\text{osc}} = \mathcal{L}_{\text{label}} + \gamma \mathcal{L}_{\text{em}}, \tag{8}$$

where $\mathcal{L}_{\text{label}}$ is the label loss, $\mathcal{L}_{\text{em}}$ is the entropy maximization loss, and $\gamma$ is a trainable parameter that balances the two objectives. The label loss is defined as:

$$\mathcal{L}_{label}(f_s(\mathbf{Z}), Y) = -\frac{1}{N_l} \sum_{i=1}^{N_l} \sum_{c=1}^{C} y_{i,c} \log(\hat{y}_{i,c}), \tag{9}$$

where $\mathbf{Z}$ denotes the embeddings of nodes in the training set obtained from Equation equation 5. $f_s(\cdot)$ is a softmax classifier consisting of a fully connected layer followed by an activation function. $N_l$ is the number of nodes from known classes, $C$ is the number of known classes, $y_{i,c}$ is the ground-truth label of the $i$-th node, and $\hat{y}_{i,c}$ is the predicted probability of assigning node $v_i$ to class $c$.

Based on the classifier outputs, a small subset of nodes, such as the bottom 10% in confidence, is selected as pseudo-unknown samples. Nodes with high confidence are assigned to known classes, whereas the selected subset exhibits relatively balanced predictions across all visible classes, mimicking the behavior of unknown-class nodes. To further enhance their distinction from known-class nodes, we perturb their embeddings with random noise:

$$\mathbf{Z}_{\text{pseudo}} = \mathbf{Z}_{\text{low-confident}} + \mathcal{N}(0, \sigma^2), \tag{10}$$

where $\mathbf{Z}_{\text{low-confident}}$ denotes the embeddings of high-entropy nodes. The perturbed nodes approximate pseudo-unknown samples.

For these pseudo-unknown nodes, we maximize their entropy using the entropy maximization loss to prevent them from being classified into any known class:

$$\mathcal{L}_{\text{em}}(f_s(\mathbf{Z}_{\text{pseudo}})) = \frac{1}{N_u} \sum_{i=1}^{N_u} \sum_{c=1}^{C} \hat{y}_{i,c} \log(\hat{y}_{i,c}), \tag{11}$$

where $N_u$ is the number of pseudo-unknown nodes.

Overall, optimizing the label loss and entropy maximization loss resembles an adversarial process. On one hand, the label loss enhances the classifier's discriminative power by aligning predictions with ground-truth labels. On the other hand, the entropy maximization loss encourages ambiguity in pseudo-unknown nodes, thereby facilitating the detection of truly unknown samples. Both losses are jointly optimized during fine-tuning via standard backpropagation.

### 3.4 Towards out-of-distribution rejection

During the testing phase, the test data are fed into the trained GNN model to obtain the output representations, which are passed through a softmax layer to produce prediction scores over $C$ classes ($P \in \mathbb{R}^{N \times C}$). For each node, we select the maximum score $\max(p_i)$ to determine whether it belongs to a known or unknown class according to:

$$\hat{y} = \begin{cases} \text{Rejection}, & \text{if } \max_{c \in \mathcal{C}} p(c \mid x_i) \leq \eta, \\ \arg\max_{c \in \mathcal{C}} p(c \mid x_i), & \text{otherwise}, \end{cases} \tag{12}$$

where $p(c|x_i)$ denotes the softmax output $f_s(\cdot)$. If all class probabilities fall below the threshold $\eta$, the node is classified as unknown; otherwise, the predicted label corresponds to the class with the highest probability.

A central challenge in open-world graph learning is determining the threshold $\eta$. To address this, we propose an automatic strategy that leverages both a validation set and pseudo-unknown nodes generated during training. Specifically, embeddings of validation nodes are passed through the GNN and softmax layer to compute their maximum prediction probabilities. The average of these values is recorded as *avg_known*. In parallel, we supplement the pseudo-unknown set with the top 10% of validation nodes exhibiting the highest entropy and compute their average maximum probability as *avg_unknown*.

The final threshold is then defined as $\eta = \frac{avg\_known + avg\_unknown}{2}$. This automatically derived threshold is subsequently applied to classify test nodes into known and unknown classes. By jointly optimizing label loss and entropy maximization loss and employing pseudo-unknown nodes for threshold estimation, our method effectively rejects nodes that do not belong to any known class.

## 4 Experiments

### 4.1 Experimental Setup

**Datasets** Following Nettack (Zügner et al., 2020) and Pro-GNN (Jin et al., 2020), we employ three widely used citation network datasets (Cora, Citeseer, and Pubmed) for node classification, considering only the largest connected component (LCC).

Table 1: Comparison of Opens-set node classification in test accuracy(%) and AUROC(%) on three citation network with one unknown class (u=1) under MetaAttack at Perturbation Rates ( 0%, 5%, 10%, 20% ). The top two performance is highlighted in bold and underline.

| | Dataset | Ptb Rate | GCN_soft | GCN_sig | GCN_soft_$\tau$ | GCN_sig_$\tau$ | OpenWGL | $\mathcal{G}^2Pxy$ | STABLE | RNCGLN | SG-GSR | OG-SNR |
|---|---|---|---|---|---|---|---|---|---|---|---|---|
| Accuracy | Cora | 0% | 50.84 | 50.58 | 80.34 | 79.43 | 80.93 | 82.92 | 74.26 | 69.34 | 80.85 | **84.22** |
| | | 5% | 49.68 | 49.55 | 68.18 | 67.40 | 74.25 | 68.56 | 63.91 | 65.98 | 78.91 | **79.04** |
| | | 10% | 48.64 | 48.77 | 64.42 | 62.74 | 70.37 | 64.29 | 73.61 | 64.17 | 76.58 | **78.35** |
| | | 20% | 47.74 | 47.74 | 61.06 | 58.73 | 67.78 | 61.83 | 69.34 | 62.23 | 77.88 | **79.82** |
| | Citeseer | 0% | 45.14 | 45.29 | 69.66 | 69.06 | 71.10 | 71.61 | 67.26 | 67.56 | **72.94** | 72.50 |
| | | 5% | 45.29 | 45.44 | 70.4 | 69.81 | 70.25 | 70.25 | 61.43 | 65.62 | 70.23 | **70.85** |
| | | 10% | 45.29 | 45.59 | 69.96 | 69.21 | 69.05 | 68.31 | 64.13 | 61.29 | 72.65 | **72.80** |
| | | 20% | 44.25 | 44.25 | 65.02 | 66.52 | 65.02 | 60.23 | 67.85 | 59.64 | 71.30 | **72.94** |
| | Pubmed | 0% | 39.99 | 39.98 | 66.58 | 64.50 | 69.65 | 57.58 | 66.39 | 64.30 | 61.96 | **79.99** |
| | | 5% | 38.68 | 38.58 | 69.82 | 69.68 | 67.13 | 56.83 | 71.30 | 62.87 | 65.20 | **78.74** |
| | | 10% | 36.56 | 36.31 | 58.50 | 57.02 | 52.60 | 56.79 | 70.47 | 62.69 | 64.41 | **77.46** |
| | | 20% | 34.18 | 33.50 | 56.79 | 56.78 | 39.89 | 56.55 | 70.27 | 62.05 | 63.88 | **77.84** |
| AUROC | Cora | 0% | 87.43 | 90.16 | 87.63 | 90.31 | 71.50 | 78.37 | 82.40 | 79.85 | 89.06 | **92.76** |
| | | 5% | 74.89 | 70.87 | 74.42 | 78.46 | 82.86 | 73.99 | 85.89 | 78.12 | 86.73 | **89.22** |
| | | 10% | 68.85 | 67.71 | 69.97 | 71.07 | 77.61 | 71.46 | 81.76 | 76.41 | 86.23 | **87.62** |
| | | 20% | 64.36 | 66.06 | 63.44 | 60.08 | 74.53 | 67.14 | 84.43 | 73.44 | 86.59 | **88.37** |
| | Citeseer | 0% | 72.76 | 72.82 | 78.33 | 77.55 | 83.56 | 69.78 | 78.44 | 79.77 | **84.61** | 84.01 |
| | | 5% | 77.25 | 75.81 | 77.29 | 78.19 | **81.20** | 69.73 | 78.76 | 80.86 | 79.86 | 80.34 |
| | | 10% | 77.10 | 78.01 | 76.55 | 78.60 | 78.22 | 73.03 | 73.65 | 79.80 | **83.87** | 80.81 |
| | | 20% | 71.42 | 75.39 | 71.46 | 74.62 | 72.38 | 70.21 | 79.85 | 67.58 | 82.50 | **83.29** |
| | Pubmed | 0% | 69.36 | 66.80 | 70.76 | 65.96 | 77.33 | 53.17 | 69.67 | 68.88 | 63.26 | **90.26** |
| | | 5% | 70.18 | 72.45 | 73.34 | 73.57 | 76.51 | 47.92 | 76.18 | 69.33 | 67.28 | **89.11** |
| | | 10% | 60.41 | 61.12 | 62.07 | 34.53 | 64.15 | 45.16 | 72.41 | 68.51 | 65.97 | **88.10** |
| | | 20% | 39.71 | 38.00 | 31.26 | 28.29 | 53.45 | 41.60 | 73.36 | 66.14 | 72.27 | **86.32** |

**Test Settings and Evaluation Metrics.** For each dataset, a subset of classes is reserved as unknow classes for testing, while the remaining classes are treated as known classes. To introduce varying levels of structural noise, we apply MetaAttack (Aburidi & Marcia, 2024) and the Random attack method, following RNCGLN (Zhu et al., 2024). We randomly sample 70% of the nodes for training, 10% for validation, and 20% for testing. Notably, nodes from the unknow classes are included exclusively in the test set. The validation set is used to determine the threshold for rejecting unknow classes. Consistent with traditional semi-supervised node classification, the entire graph is utilized for model training on each dataset.

**Implementation Details.** Generally, OG-SNR employs GCN (Kipf & Welling, 2016b) as the backbone for experimental evaluation, unless stated otherwise. The GCN architecture consists of two hidden layers with dimensions 512 and 128, followed by a fully connected layer of size 64. For the hyperparameters, We set $\epsilon = 0.3$ and $\sigma = 0.2$. OG-SNR is implemented in PyTorch and optimized using stochastic gradient descent with a learning rate of $1 \times 10^{-3}$.

Baseline methods are evaluated according to the configurations reported in their original papers, using identical parameters unless specified otherwise, and the best results are reported. For each experiment, both the baselines and the proposed method are applied to the same training, validation, and testing sets. Hyperparameters are tuned to achieve optimal performance on the validation set.

**Baselines.** To validate the effectiveness of OG-SNR, three categories of baselines are included in our experiments:

- 1) Closed-set classification methods: *GCN_soft* and *GCN_sig*. These methods utilize GCN (Kipf & Welling, 2016b) with different output layers: *GCN_soft* employs a softmax layer, while *GCN_sig* uses multiple 1-vs-rest sigmoid layers. They lack the ability to recognize unknown classes.
- 2) Open-set classification methods: *GCN_soft_$\tau$*, *GCN_sig_$\tau$*, *OpenWGL* (Wu et al., 2020), and $\mathcal{G}^2Pxy$ (Zhang et al., 2023) are commonly used baselines for open-set node classification. Specifically, *GCN_soft_$\tau$* and *GCN_sig_$\tau$* are extensions of *GCN_soft* and *GCN_sig*, which apply a probability threshold selected from $\{0.1, 0.2, \ldots, 0.9\}$ to perform open-set recognition. If all predicted probabilities for a sample fall below the threshold, the sample is rejected as belonging to an unknown class; otherwise, the class with the highest probability is assigned as the prediction. *OpenWGL* and $\mathcal{G}^2Pxy$ are representative open-set graph learning methods. *OpenWGL* introduces an uncertainty loss based on graph reconstruction of unlabeled data and employs an adaptive threshold to detect unknown-class samples. $\mathcal{G}^2Pxy$ is a generative approach that constructs pseudo-unknown nodes, transforming a closed-set classifier into an open-set one. However, both methods exhibit limited robustness, particularly under structural noise or adversarial perturbations.

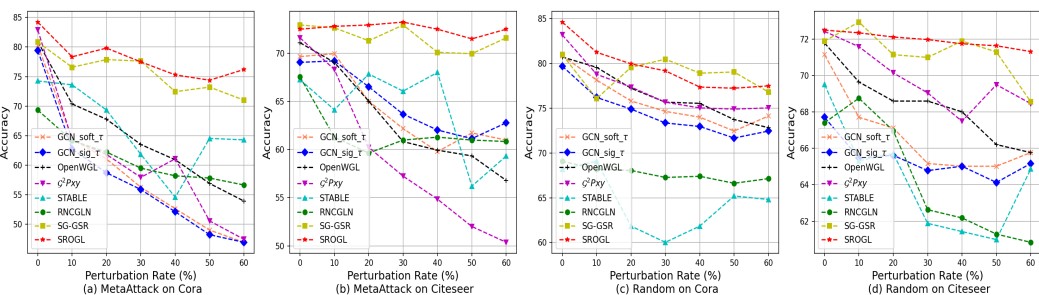

Figure 2: Cora and Citeseer with one unknown class (u=1) under MetaAttack and Random attack

- 3) Graph Structure Learning methods: *STABLE* (Li et al., 2022b), *RNCGLN* (Zhu et al., 2024), and *SG-GSR* (In et al., 2024) are representative methods for robust graph learning. *STABLE* is a contrastive learning approach that leverages robustness-oriented augmentations to obtain node representations for structure refinement. This method effectively captures structural information and demonstrates insensitivity to perturbations. *RNCGLN* jointly addresses label noise and structural noise. It employs graph contrastive loss and self-attention mechanisms for learning, and uses pseudo-graphs and pseudo-labels to mitigate structural and label noise, respectively. *SG-GSR* introduces a self-guided graph structure optimization framework that extracts a clean subgraph and trains a robust node classifier via graph augmentation and group training strategies.

## 4.2 RESULT ANALYSIS OF OPEN-SET NODE CLASSIFICATION WITH EDGE NOISE

We compare our method with all baseline approaches for node classification under two types of edge attacks on three datasets. Table 1 presents results for edge noise proportions ranging from 0% to 20% under MetaAttack. Only Accuracy and AUROC metrics are displayed here. Results for Random attack and additional experimental details are provided in the Appendix.

Our method consistently achieves the best performance in open-set node classification across all noise ratios, followed by SG-GSR, STABLE, OpenWGL, RNCGLN, GCN_soft_$\tau$, GCN_sig_$\tau$, $\mathcal{G}^2Pxy$, GCN_soft, and GCN_sig. Compared to the strongest baseline (SG-GSR), our approach achieves an average improvement of 5.64% in Accuracy and 7.85% in AUROC under the MetaAttack setting. Under Random attack, the average gains are 3.52% in Accuracy and 8.41% in AUROC (see Appendix A.2).

We further analyzed classification accuracy for known and unknown classes separately. Compared to closed-set classifiers, OG-SNR sacrifices only a small portion of known-class accuracy (from 80.39% for GCN_soft to 76.12%), while substantially improving unknown-class detection, achieving an average accuracy of 75.68%. Compared with other open-set methods such as OpenWGL, OG-SNR achieves average improvements of 19.22%, 17.07%, and 18.21% in known-class accuracy, unknown-class accuracy, and overall accuracy, respectively. The F1 score is improved by an average of 18.18%. Detailed results are provided in the Appendix A.2.

## 4.3 RESULT ANALYSIS OF NOISE ROBUSTNESS

We investigate the sensitivity of all methods to different noise ratios to analyze their robustness. Specifically, we vary the noise ratio across $\{0, 0.1, 0.2, 0.3, 0.4, 0.5, 0.6\}$. Methods GCN_soft and GCN_sig are excluded from this experiment due to their poor overall performance and lack of open-set capability. All results are presented in Figure 2.

Across the four figures depicting graph noise variations, all methods exhibit a decreasing performance trend as noise increases. However, our method consistently outperforms the baselines and demonstrates greater stability. For instance, in the Citeseer dataset (Figure 2(d)), the performance of our method remains nearly constant across different graph noise levels, whereas the best comparison method, SG-GSR, experiences a performance drop of 3.34% as graph noise increases from 0 to 0.6. On average, our method outperforms SG-GSR by 0.7% across these noise ratios.

Overall, our method exhibits the most stable performance under various perturbation rates of different noise types, including MetaAttack and Random attack. This robustness can be attributed to the graph structure refinement module, which mitigates the effects of structural noise, the curriculum-

Table 2: Ablation study of open-set node classification in test accuracy(%) and AUROC(%) on three citation network with one unknown class (u=1) under MetaAttack at Perturbation Rate 20%.

| Methods | Cora | | Citeseer | | Pubmed | |
|---|---|---|---|---|---|---|
| | Acc. | AUROC | Acc. | AUROC | Acc. | AUROC |
| OG-SNR¬em | 68.95 | 77.82 | 66.82 | 74.98 | 58.84 | 58.26 |
| OG-SNR¬r | 76.33 | 84.81 | 72.80 | 82.52 | 76.45 | 85.71 |
| OG-SNR¬c | 76.58 | 86.93 | 68.76 | 79.75 | 75.15 | 82.41 |
| OG-SNR | **79.82** | **88.37** | **72.94** | **83.29** | **77.84** | **86.32** |

based GNN training, and the combination of entropy maximization loss with automatic thresholding, enabling effective out-of-distribution detection. In contrast, other methods typically address only one of these factors.

### 4.4 ABLATION STUDY

We conduct an ablation study to assess the contribution of each main component and setting in OG-SNR:

- OG-SNR¬em: a variant without the entropy maximization loss and the automatically determined threshold. In this variant, the optimal threshold for OOD detection is manually selected from the range $\{0.1, 0.2, \ldots, 0.9\}$.

- OG-SNR¬r: a variant without the edge homophily-based graph structure refinement module.

- OG-SNR¬c: a variant that treats all edges equally, without employing the curriculum learning framework during training.

Table 2 presents the performance of OG-SNR and its three variants. The results demonstrate that the graph structure refinement module, the entropy maximization loss with automatic thresholding, and the curriculum learning framework each play a critical role. Graph structure refinement eliminates easily detectable noisy edges, entropy maximization with automatic thresholding facilitates distinguishing unknown from known nodes, and the curriculum learning framework emphasizes simpler edges, thereby enhancing robustness and mitigating the impact of noisy edges.

### 4.5 PARAMETER SENSITIVITY ANALYSIS

In our parameter analysis, we investigated the sensitivity of $k$ and $\lambda$. Here, $k$ controls the number of edges added in the Graph Structure Refinement module, and $\lambda$ specifies the number of edges added per iteration in the curriculum learning framework. Experiments on the Cora dataset under a 20% perturbation rate with MetaAttack indicate that adding edges consistently improves performance regardless of $k$. As $\lambda$ increases, accuracy initially rises and then declines, reaching a peak at $\lambda = 3$. Small values of $\lambda$ limit the effectiveness of curriculum

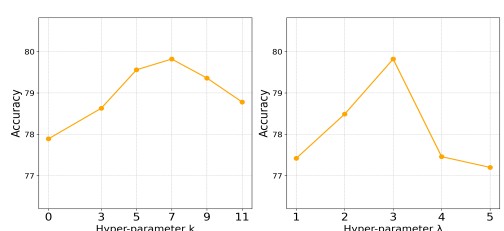

Figure 3: Parameter sensitivity analysis on Cora.

learning, whereas large values may result in overly aggressive updates and suboptimal performance.

### 5 CONCLUSIONS

This paper proposes a structure-noise-robust method for open-set node classification. The approach refines graph structures using expressive node embeddings and incorporates a curriculum learning paradigm to improve model robustness. Furthermore, by generating pseudo-OOD nodes under the constraint of an entropy maximization loss, the method achieves superior performance. To the best of our knowledge, this is the first study to address this problem.

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

# A APPENDIX

## A.1 RELATED WORK

**Open-set recognition on graphs.** In open-set recognition, the goal is not only to assign instances to known classes but also to identify and reject instances that belong to unknown classes. Past research has focused on open-set recognition on the image data (Vareto et al., 2017; Yoshihashi et al., 2019; Oza & Patel, 2019; Baktashmotlagh et al., 2018; Geng et al., 2020) and text data (Prakhya et al., 2017; Doan & Kalita, 2017); however, in recent years, researchers have begun to extend the problem to graph data (Zhang et al., 2023; 2024b; Guo et al., 2023; Li et al., 2022a; Wu et al., 2021; 2020). For example, $G^2Pxy$ (Zhang et al., 2023) generates pseudo unknown class nodes to transform a closed-set classifier into an open-set one. OpenWGL (Wu et al., 2020) proposes an uncertain node representation learning principle to test a classifier's response on a node, which can differentiate whether a node belongs to the unknow class. NGC (Wu et al., 2021) transforms image instances to graph through k-NN, and utilizes the distances between the instance and the classes center and a threshold to identify unknown classes. PGL (Luo et al., 2020) extends a previous unsupervised domain adaption framework for the open-set scenario with graph neural networks. However, these methods rely on clean graph structures and do not account for robustness to structured noise in open-set scenarios.

**Graph structure learning.** Learning from noisy graph structures is a classic problem in the field of graph structure learning, with numerous studies already conducted in this area (Li et al., 2024; Zhiyao et al., 2024; Lin et al., 2024; Guo et al., 2024; In et al., 2024; Zhu et al., 2024). Extensive research has shown that GNNs are highly vulnerable to adversarial attacks (Dai et al., 2018; Wu et al., 2019; Zhu et al., 2022b; Zügner et al., 2018; 2020). Attackers can significantly degrade the performance of GNNs by making limited modifications to graph data (i.e., structure and features). To enhance the robustness of GNN models, various methods have been proposed, including graph structure refinement (Dai et al., 2022; Jin et al., 2020; In et al., 2024), adversarial training (Kong et al., 2022), robust node representations (In et al., 2023; Zhu et al., 2019), novel message-passing schemes (Lei et al., 2022; Liu et al., 2021a), and leveraging low-rank components of graphs (Dai et al., 2022). Among these methods, the most relevant to our work is Graph Structure Refinement, which aims to learn an improved graph structure from a given graph and has recently been utilized to mitigate the impact of adversarial edges in attacked graphs. ProGNN (Jin et al., 2020) refines attacked graph structures by satisfying multiple real-world graph properties, such as feature smoothness, sparsity, and low-rankness. SG-GSR (In et al., 2024) replaces attacked graph structures with clean subgraphs and refines the attacked graph structure based on multi-aspect information. RNCGLN (Zhu et al.,

2024) simultaneously addresses graph structural noise and label noise by using graph contrastive loss for local graph learning and self-attention for global graph learning.

However, the aforementioned methods either focus solely on detecting out-of-distribution (OOD) data or exclusively on addressing graph structural noise, failing to consider both issues simultaneously. Furthermore, directly applying graph structure noise-resistant methods to OOD detection has not yielded satisfactory results. In this work, we introduce a novel framework that simultaneously addresses both OOD detection and graph structural noise problems.

## A.2 ADDITIONAL EXPERIMENTAL RESULTS

In this section, we provide more detailed experimental results.

Table 3: Comparison of Opens-set node classification in test accuracy(%) and AUROC(%) on three citation network with one unknown class (u=1) under random attack at Perturbation Rates (0%, 5%, 10%, 20%). The top two performance is highlighted in bold and underline.

| | Dataset | Ptb Rate | GCN_soft | GCN_sig | GCN_soft_$\tau$ | GCN_sig_$\tau$ | OpenWGL | $\mathcal{G}^2Pxy$ | STABLE | RNCGLN | SG-GSR | ours |
|---|---|---|---|---|---|---|---|---|---|---|---|---|
| Accuracy | Cora | 0% | 50.97 | 50.58 | 80.85 | 79.69 | 80.72 | 83.18 | 61.19 | 69.08 | 80.98 | **84.61** |
| | | 5% | 50.45 | 50.32 | 77.75 | 76.97 | 80.33 | 80.46 | 69.34 | 68.05 | 80.34 | **80.72** |
| | | 10% | 50.06 | 50.32 | 78.14 | 76.20 | 79.56 | 78.78 | 69.08 | 68.18 | 76.07 | **81.24** |
| | | 20% | 49.29 | 49.03 | 75.81 | 74.90 | 77.23 | 77.36 | 61.84 | 68.05 | 79.56 | **79.95** |
| | Citeseer | 0% | 45.14 | 45.29 | 71.15 | 67.71 | 71.79 | 72.43 | 69.51 | 67.41 | 71.90 | **72.50** |
| | | 5% | 45.14 | 44.84 | 69.36 | 67.12 | 70.85 | 70.49 | 66.82 | 67.71 | 71.56 | **72.65** |
| | | 10% | 44.84 | 45.14 | 67.71 | 65.47 | 69.65 | 71.59 | 56.20 | 68.76 | **72.94** | 72.35 |
| | | 20% | 44.69 | 44.25 | 67.12 | 65.62 | 68.60 | 70.15 | 65.72 | 66.97 | 71.15 | **72.12** |
| | Pubmed | 0% | 40.01 | 39.95 | 66.26 | 68.70 | 69.89 | 57.62 | 67.66 | 64.21 | 58.37 | **80.49** |
| | | 5% | 39.57 | 39.58 | 66.57 | 67.99 | 68.21 | 57.45 | 67.51 | 64.06 | 60.42 | **78.16** |
| | | 10% | 39.37 | 39.25 | 64.84 | 63.96 | 66.63 | 57.55 | 69.49 | 63.58 | 63.74 | **76.52** |
| | | 20% | 39.09 | 38.92 | 67.01 | 65.43 | 63.70 | 57.23 | 70.87 | 62.77 | 61.22 | **76.18** |
| AUROC | Cora | 0% | 88.77 | 89.50 | 88.43 | 89.84 | **90.95** | 78.45 | 79.53 | 79.75 | 89.07 | 82.53 |
| | | 5% | 85.85 | 87.82 | 87.28 | 88.63 | **90.14** | 77.30 | 80.98 | 78.81 | 88.96 | 89.26 |
| | | 10% | 85.96 | 87.16 | 85.95 | 87.98 | 88.93 | 76.95 | 83.10 | 79.05 | 84.91 | **89.40** |
| | | 20% | 83.15 | 85.99 | 83.69 | 84.79 | 87.44 | 74.12 | 76.70 | 78.65 | **88.77** | 87.53 |
| | Citeseer | 0% | 76.13 | 75.86 | 77.42 | 76.33 | 83.56 | 70.04 | 80.65 | 79.94 | 84.06 | **84.26** |
| | | 5% | 75.23 | 73.12 | 76.97 | 75.57 | 81.91 | 70.07 | 78.07 | 78.73 | 81.26 | **82.84** |
| | | 10% | 73.19 | 74.42 | 78.05 | 74.10 | 80.86 | 70.58 | 79.76 | 78.65 | **83.55** | 81.92 |
| | | 20% | 71.39 | 72.96 | 72.76 | 73.38 | 79.09 | 68.09 | 79.78 | 76.46 | 81.71 | **84.79** |
| | Pubmed | 0% | 67.81 | 67.74 | 69.43 | 69.28 | 77.35 | 53.17 | 72.82 | 68.89 | 55.85 | **90.40** |
| | | 5% | 70.35 | 67.79 | 70.96 | 68.82 | 76.42 | 54.95 | 69.12 | 68.37 | 59.70 | **89.45** |
| | | 10% | 70.24 | 68.58 | 67.83 | 67.35 | 75.65 | 54.56 | 74.20 | 67.77 | 67.60 | **88.23** |
| | | 20% | 69.91 | 69.72 | 70.67 | 68.81 | 72.56 | 53.50 | 75.43 | 67.01 | 63.97 | **87.52** |

Table 4: Comparison of Open-Set Node Classification: Test Accuracy (%), F1 Score (%), and AUROC (%) for Known Class Classification and Unknown Class Recognition on Three Citation Networks with One Unknown Class (u=1) Under MetaAttack at Perturbation Rates (0%, 10%, 20%, 30%)

| Ptb Rate | Methods | Cora | | | | | Citeseer | | | | | Pubmed | | | | |
|---|---|---|---|---|---|---|---|---|---|---|---|---|---|---|---|---|
| | | tacc | F1 | auroc | kacc | uacc | tacc | F1 | auroc | kacc | uacc | tacc | F1 | auroc | kacc | uacc |
| 0% | GCN_soft | 50.84 | 55.62 | 87.43 | 91.61 | 0.00 | 45.14 | 42.74 | 72.76 | 83.66 | 0.00 | 39.99 | 38.66 | 69.36 | 92.54 | 0.00 |
| | GCN_sig | 50.58 | 54.55 | 90.16 | 91.14 | 0.00 | 45.29 | 47.06 | 72.82 | 83.93 | 0.00 | 39.98 | 38.82 | 66.80 | 92.51 | 0.00 |
| | GCN_soft_$\tau$ | 80.34 | 76.84 | 87.63 | 76.69 | 84.88 | 69.66 | 63.14 | 78.33 | 65.37 | 74.68 | 66.58 | 62.98 | 70.76 | 53.73 | 76.36 |
| | GCN_sig_$\tau$ | 79.43 | 76.36 | 90.31 | 73.66 | 86.63 | 69.06 | 59.16 | 77.55 | 64.54 | 74.35 | 64.50 | 61.54 | 65.96 | 54.02 | 72.48 |
| | OpenWGL | 80.93 | 77.11 | 71.50 | 73.19 | 90.69 | 71.10 | 60.08 | 83.56 | 63.15 | 84.09 | 69.65 | 69.58 | 77.33 | 77.90 | 63.37 |
| | $\mathcal{G}^2Pxy$ | 82.92 | 81.84 | 78.37 | 85.31 | 79.94 | 71.61 | 62.32 | 69.78 | 72.85 | 72.22 | 57.58 | 39.39 | 53.17 | 17.03 | 88.44 |
| | STABLE | 74.26 | 54.90 | 82.40 | 65.73 | 84.88 | 67.26 | 54.17 | 78.44 | 63.99 | 71.10 | 66.39 | 65.53 | 69.67 | 65.45 | 67.10 |
| | RNCGLN | 69.34 | 59.07 | 79.85 | 72.49 | 65.41 | 67.56 | 62.09 | 79.77 | 79.78 | 53.25 | 64.30 | 63.81 | 68.88 | 68.24 | 61.30 |
| | SG-GSR | 80.85 | 75.58 | 89.06 | 72.26 | 91.57 | 72.94 | 62.43 | 84.61 | 71.47 | 74.68 | 61.96 | 58.76 | 63.26 | 51.20 | 70.14 |
| | ours | 84.22 | 81.93 | 92.76 | 82.75 | 86.05 | 72.50 | 65.46 | 84.01 | 72.85 | 72.08 | 79.99 | 79.27 | 90.26 | 88.57 | 73.46 |
| 10% | GCN_soft | 48.64 | 54.35 | 68.85 | 87.65 | 0.00 | 45.29 | 43.81 | 77.10 | 83.93 | 0.00 | 36.56 | 34.41 | 60.41 | 84.60 | 0.00 |
| | GCN_sig | 48.77 | 54.83 | 67.71 | 87.88 | 0.00 | 45.59 | 43.68 | 78.01 | 84.49 | 0.00 | 36.31 | 34.68 | 61.12 | 84.02 | 0.00 |
| | GCN_soft_$\tau$ | 64.42 | 66.47 | 69.97 | 72.03 | 54.94 | 69.96 | 59.26 | 76.55 | 68.98 | 71.10 | 58.50 | 56.18 | 62.07 | 53.06 | 62.64 |
| | GCN_sig_$\tau$ | 62.74 | 61.89 | 71.07 | 73.43 | 49.42 | 69.21 | 57.86 | 78.60 | 65.93 | 73.05 | 57.02 | 26.71 | 34.53 | 2.08 | 98.83 |
| | OpenWGL | 70.37 | 70.63 | 77.61 | 72.49 | 67.73 | 69.05 | 56.56 | 78.22 | 60.66 | 78.89 | 52.60 | 54.15 | 64.15 | 67.72 | 41.09 |
| | $\mathcal{G}^2Pxy$ | 64.29 | 68.62 | 71.46 | 81.58 | 42.73 | 68.31 | 58.27 | 73.03 | 78.39 | 56.49 | 56.79 | 24.31 | 45.16 | 00.12 | 99.92 |
| | STABLE | 73.61 | 62.21 | 81.76 | 68.30 | 80.23 | 64.13 | 46.71 | 73.65 | 52.35 | 77.92 | 70.47 | 66.43 | 72.41 | 56.80 | 80.87 |
| | RNCGLN | 64.17 | 55.51 | 76.41 | 69.93 | 56.98 | 61.29 | 56.14 | 79.80 | 81.44 | 37.66 | 62.69 | 62.09 | 68.51 | 70.19 | 56.98 |
| | SG-GSR | 76.58 | 74.86 | 86.23 | 73.66 | 81.58 | 72.65 | 64.99 | 83.87 | 74.79 | 70.13 | 69.65 | 65.97 | 65.97 | 47.55 | 77.24 |
| | ours | 78.35 | 75.10 | 87.62 | 75.52 | 82.27 | 72.80 | 62.29 | 80.81 | 78.39 | 66.23 | 77.46 | 76.26 | 88.10 | 84.25 | 72.29 |
| 20% | GCN_soft | 47.74 | 54.75 | 64.36 | 86.01 | 0.00 | 44.25 | 41.60 | 71.42 | 81.99 | 0.00 | 34.18 | 34.31 | 39.71 | 79.09 | 0.00 |
| | GCN_sig | 47.74 | 55.77 | 66.06 | 86.01 | 0.00 | 44.25 | 42.58 | 75.39 | 81.99 | 0.00 | 33.50 | 34.00 | 38.00 | 77.52 | 0.00 |
| | GCN_soft_$\tau$ | 61.06 | 65.89 | 63.44 | 78.55 | 39.24 | 65.02 | 54.36 | 71.46 | 65.37 | 64.61 | 56.79 | 24.23 | 31.26 | 0.06 | 99.98 |
| | GCN_sig_$\tau$ | 58.73 | 63.50 | 60.08 | 73.43 | 40.41 | 66.52 | 55.55 | 74.62 | 63.99 | 69.48 | 56.78 | 24.14 | 28.29 | 0.00 | 100.00 |
| | OpenWGL | 67.78 | 69.16 | 74.53 | 72.49 | 61.91 | 65.02 | 52.64 | 72.38 | 57.06 | 74.35 | 39.89 | 37.69 | 53.45 | 41.05 | 38.99 |
| | $\mathcal{G}^2Pxy$ | 61.83 | 66.18 | 67.14 | 80.65 | 38.37 | 60.23 | 52.79 | 70.21 | 75.90 | 41.88 | 56.55 | 24.57 | 41.60 | 00.38 | 99.31 |
| | STABLE | 69.34 | 69.03 | 84.43 | 80.42 | 55.52 | 67.85 | 59.92 | 79.85 | 59.12 | 69.16 | 70.27 | 67.22 | 73.36 | 61.35 | 77.07 |
| | RNCGLN | 62.23 | 55.03 | 73.44 | 69.93 | 52.62 | 59.64 | 49.91 | 67.58 | 55.68 | 64.29 | 62.05 | 60.39 | 66.14 | 64.14 | 60.47 |
| | SG-GSR | 77.88 | 73.53 | 86.59 | 74.59 | 81.98 | 71.30 | 63.62 | 82.50 | 74.52 | 67.53 | 63.88 | 56.21 | 72.27 | 43.07 | 79.72 |
| | ours | 79.82 | 73.64 | 88.37 | 77.39 | 78.49 | 72.94 | 64.34 | 83.29 | 68.70 | 77.92 | 77.84 | 75.88 | 86.32 | 75.18 | 79.87 |
| 30% | GCN_soft | 46.70 | 53.64 | 59.33 | 84.15 | 0.00 | 43.35 | 38.31 | 72.31 | 80.33 | 0.00 | 31.84 | 32.81 | 27.32 | 73.68 | 0.00 |
| | GCN_sig | 46.57 | 53.96 | 57.32 | 83.92 | 0.00 | 43.65 | 39.86 | 73.03 | 80.89 | 0.00 | 31.91 | 32.98 | 26.70 | 73.84 | 0.00 |
| | GCN_soft_$\tau$ | 56.27 | 61.71 | 58.30 | 72.73 | 35.76 | 62.18 | 50.07 | 71.72 | 55.40 | 70.13 | 56.78 | 24.14 | 23.21 | 0.00 | 100.00 |
| | GCN_sig_$\tau$ | 55.89 | 61.94 | 58.27 | 75.06 | 31.98 | 63.68 | 51.48 | 74.30 | 59.56 | 68.51 | 56.78 | 24.14 | 20.33 | 0.00 | 100.00 |
| | OpenWGL | 63.51 | 64.38 | 70.63 | 67.13 | 59.01 | 60.83 | 49.96 | 69.46 | 57.06 | 65.25 | 37.66 | 35.90 | 52.58 | 39.32 | 36.41 |
| | $\mathcal{G}^2Pxy$ | 57.95 | 61.31 | 64.24 | 77.38 | 33.72 | 57.24 | 50.67 | 66.88 | 77.00 | 34.09 | 55.77 | 25.21 | 39.01 | 01.12 | 97.36 |
| | STABLE | 61.84 | 42.69 | 64.02 | 50.35 | 76.16 | 66.07 | 58.98 | 80.97 | 76.18 | 54.22 | 68.25 | 63.28 | 65.38 | 51.59 | 80.94 |
| | RNCGLN | 59.51 | 53.67 | 71.02 | 69.70 | 46.80 | 60.99 | 49.52 | 67.31 | 52.63 | 70.78 | 62.32 | 59.24 | 65.11 | 59.11 | 64.76 |
| | SG-GSR | 77.62 | 73.70 | 85.18 | 74.59 | 81.40 | 72.94 | 64.73 | 83.19 | 77.29 | 67.86 | 56.78 | 24.14 | 48.99 | 0.00 | 100.00 |
| | ours | 77.49 | 74.90 | 86.59 | 75.29 | 80.23 | 73.24 | 64.27 | 82.79 | 66.76 | 80.84 | 75.45 | 74.27 | 88.19 | 83.29 | 69.49 |

Table 5: Comparison of Open-Set Node Classification: Test Accuracy (%), F1 Score (%), and AUROC (%) for Known Class Classification and Unknown Class Recognition on Three Citation Networks with One Unknown Class (u=1) Under MetaAttack at Perturbation Rates ( 40%, 50%, 60%)

| Ptb Rate | Methods | Cora | | | | | Citeseer | | | | | Pubmed | | | | |
|---|---|---|---|---|---|---|---|---|---|---|---|---|---|---|---|---|
| | | tacc | F1 | auroc | kacc | uacc | tacc | F1 | auroc | kacc | uacc | tacc | F1 | auroc | kacc | uacc |
| 40% | GCN_soft | 45.67 | 52.69 | 51.15 | 82.28 | 0.00 | 42.45 | 41.08 | 55.81 | 78.67 | 0.00 | 30.51 | 31.64 | 22.11 | 70.61 | 0.00 |
| | GCN_sig | 45.28 | 52.27 | 51.49 | 81.59 | 0.00 | 42.60 | 37.31 | 71.06 | 78.95 | 0.00 | 30.67 | 31.86 | 23.03 | 70.96 | 0.00 |
| | GCN_soft_$\tau$ | 52.65 | 57.49 | 51.97 | 68.53 | 32.85 | 59.79 | 49.55 | 68.57 | 63.43 | 55.52 | 56.78 | 24.14 | 17.48 | 0.00 | 100.00 |
| | GCN_sig_$\tau$ | 52.13 | 58.46 | 50.45 | 71.33 | 28.20 | 62.03 | 48.16 | 67.92 | 59.28 | 65.26 | 61.96 | 41.65 | 81.80 | 24.02 | 90.84 |
| | OpenWGL | 60.93 | 63.19 | 69.81 | 67.83 | 52.32 | 59.94 | 46.30 | 67.90 | 49.58 | 72.07 | 36.19 | 35.25 | 51.12 | 39.77 | 33.46 |
| | $\mathcal{G}^2Pxy$ | 61.09 | 65.05 | 61.51 | 78.125 | 31.81 | 54.85 | 49.06 | 62.15 | 77.28 | 28.57 | 54.87 | 25.56 | 37.97 | 01.72 | 95.32 |
| | STABLE | 54.59 | 58.03 | 61.30 | 75.52 | 28.48 | 68.01 | 57.26 | 78.59 | 71.75 | 63.64 | 64.77 | 55.47 | 55.43 | 38.68 | 84.62 |
| | RNCGLN | 58.21 | 52.73 | 68.97 | 70.16 | 43.31 | 61.29 | 49.31 | 67.60 | 51.25 | 73.05 | 61.42 | 58.78 | 64.24 | 60.07 | 62.44 |
| | SG-GSR | 72.45 | 68.48 | 83.21 | 70.86 | 74.42 | 70.10 | 59.34 | 81.06 | 65.65 | 75.32 | 56.78 | 24.14 | 32.37 | 0.00 | 100.00 |
| | ours | 75.29 | 74.04 | 85.95 | 75.76 | 74.71 | 72.50 | 62.06 | 83.38 | 68.42 | 77.27 | 75.60 | 74.37 | 87.98 | 83.03 | 69.95 |
| 50% | GCN_soft | 44.24 | 51.76 | 47.03 | 79.72 | 0.00 | 42.60 | 39.76 | 62.53 | 78.95 | 0.00 | 29.34 | 30.47 | 19.99 | 67.88 | 0.00 |
| | GCN_sig | 44.11 | 51.82 | 45.49 | 79.49 | 0.00 | 42.15 | 37.69 | 60.30 | 78.12 | 0.00 | 29.46 | 30.62 | 20.26 | 68.17 | 0.00 |
| | GCN_soft_$\tau$ | 49.03 | 56.41 | 48.83 | 70.63 | 22.09 | 61.73 | 45.19 | 69.88 | 51.80 | 73.38 | 56.78 | 24.14 | 20.60 | 0.00 | 100.00 |
| | GCN_sig_$\tau$ | 48.25 | 56.80 | 46.21 | 72.96 | 17.44 | 61.14 | 48.34 | 72.08 | 59.00 | 63.64 | 56.78 | 24.14 | 17.48 | 0.00 | 100.00 |
| | OpenWGL | 56.92 | 58.16 | 64.94 | 63.86 | 48.25 | 59.34 | 45.19 | 68.04 | 45.15 | 75.97 | 36.5 | 34.34 | 51.11 | 36.75 | 36.31 |
| | $\mathcal{G}^2Pxy$ | 50.58 | 48.62 | 55.23 | 69.46 | 27.03 | 52.01 | 46.97 | 59.77 | 76.45 | 23.37 | 54.16 | 26.17 | 37.37 | 2.62 | 93.39 |
| | STABLE | 64.55 | 42.72 | 71.18 | 51.05 | 81.48 | 56.20 | 49.65 | 68.30 | 70.91 | 38.96 | 62.84 | 46.95 | 47.33 | 25.49 | 91.27 |
| | RNCGLN | 57.83 | 52.33 | 67.45 | 68.30 | 44.77 | 60.99 | 49.11 | 67.74 | 50.42 | 73.38 | 61.67 | 58.43 | 63.39 | 58.34 | 64.20 |
| | SG-GSR | 73.22 | 67.53 | 82.29 | 61.77 | 87.50 | 69.96 | 59.77 | 81.58 | 72.02 | 67.53 | 56.78 | 24.14 | 28.05 | 0.00 | 100.00 |
| | ours | 74.39 | 72.22 | 84.86 | 72.03 | 77.33 | 71.75 | 65.57 | 80.70 | 70.91 | 72.73 | 73.78 | 72.65 | 86.69 | 82.68 | 67.00 |
| 60% | GCN_soft | 42.69 | 49.73 | 44.27 | 76.92 | 0.00 | 42.30 | 41.59 | 56.60 | 78.39 | 0.00 | 28.37 | 29.54 | 19.29 | 65.64 | 0.00 |
| | GCN_sig | 42.95 | 50.30 | 43.03 | 77.39 | 0.00 | 41.85 | 37.67 | 61.24 | 77.56 | 0.00 | 28.49 | 29.73 | 18.17 | 65.93 | 0.00 |
| | GCN_soft_$\tau$ | 46.96 | 55.15 | 41.83 | 70.40 | 17.73 | 60.99 | 42.57 | 66.64 | 49.31 | 74.68 | 57.68 | 28.82 | 77.58 | 3.91 | 98.61 |
| | GCN_sig_$\tau$ | 46.96 | 55.67 | 42.60 | 69.70 | 18.60 | 62.78 | 47.23 | 71.84 | 54.85 | 72.08 | 56.78 | 24.14 | 18.27 | 0.00 | 100.00 |
| | OpenWGL | 53.94 | 53.99 | 61.53 | 61.53 | 44.76 | 56.80 | 43.26 | 66.62 | 45.15 | 70.45 | 35.80 | 33.70 | 51.60 | 36.15 | 35.53 |
| | $\mathcal{G}^2Pxy$ | 47.47 | 45.70 | 54.04 | 67.83 | 22.09 | 50.37 | 45.80 | 58.80 | 75.34 | 21.10 | 53.27 | 26.37 | 37.50 | 03.23 | 91.37 |
| | STABLE | 64.29 | 57.62 | 80.68 | 70.63 | 56.40 | 59.34 | 40.71 | 68.83 | 64.29 | 65.26 | 61.32 | 40.10 | 28.10 | 17.71 | 94.52 |
| | RNCGLN | 56.66 | 59.98 | 65.18 | 75.76 | 32.85 | 60.84 | 49.08 | 68.11 | 49.86 | 73.70 | 61.14 | 58.10 | 62.79 | 58.85 | 62.88 |
| | SG-GSR | 71.02 | 68.51 | 80.71 | 69.70 | 72.67 | 71.60 | 60.33 | 81.89 | 67.04 | 76.95 | 55.16 | 43.99 | 48.94 | 25.23 | 77.94 |
| | ours | 76.20 | 72.34 | 85.11 | 70.63 | 83.14 | 72.50 | 60.32 | 82.29 | 65.65 | 80.52 | 73.21 | 71.85 | 86.21 | 80.56 | 67.61 |

Table 6: Comparison of Open-Set Node Classification: Test Accuracy (%), F1 Score (%), and AUROC (%) for Known Class Classification and Unknown Class Recognition on Three Citation Networks with One Unknown Class (u=1) Under random Attack at Perturbation Rates (0%, 10%, 20%, 30%)

| Ptb Rate | Methods | Cora | | | | | Citeseer | | | | | Pubmed | | | | |
|---|---|---|---|---|---|---|---|---|---|---|---|---|---|---|---|---|
| | | tacc | F1 | auroc | kacc | uacc | tacc | F1 | auroc | kacc | uacc | tacc | F1 | auroc | kacc | uacc |
| 0% | GCN_soft | 50.97 | 55.65 | 88.77 | 91.84 | 0.00 | 45.14 | 42.65 | 76.13 | 83.66 | 0.00 | 40.01 | 38.90 | 67.81 | 92.57 | 0.00 |
| | GCN_sig | 50.58 | 57.14 | 89.50 | 91.14 | 0.00 | 45.29 | 44.54 | 75.86 | 83.93 | 0.00 | 39.95 | 38.62 | 67.74 | 92.44 | 0.00 |
| | GCN_soft_$\tau$ | 80.85 | 77.13 | 88.43 | 73.66 | 89.83 | 71.15 | 65.76 | 77.42 | 66.76 | 76.30 | 66.26 | 62.61 | 69.43 | 53.47 | 75.99 |
| | GCN_sig_$\tau$ | 79.69 | 76.72 | 89.84 | 77.39 | 82.56 | 67.71 | 61.32 | 76.33 | 70.64 | 64.29 | 68.70 | 56.94 | 69.28 | 44.44 | 87.16 |
| | OpenWGL | 80.72 | 78.23 | 90.95 | 72.72 | 90.69 | 71.79 | 60.08 | 83.56 | 63.15 | 84.09 | 69.89 | 69.78 | 77.35 | 77.65 | 63.98 |
| | $\mathcal{G}^2Pxy$ | 83.18 | 81.83 | 78.45 | 85.31 | 80.52 | 72.43 | 62.23 | 70.04 | 73.14 | 71.29 | 57.62 | 39.41 | 53.17 | 17.03 | 88.52 |
| | STABLE | 72.54 | 56.01 | 79.53 | 62.70 | 75.00 | 69.51 | 55.43 | 80.65 | 61.50 | 78.90 | 67.66 | 56.88 | 72.82 | 39.96 | 88.74 |
| | RNCGLN | 69.08 | 58.78 | 79.75 | 72.03 | 65.41 | 67.41 | 61.80 | 79.94 | 79.78 | 52.92 | 64.21 | 63.83 | 68.89 | 69.36 | 60.30 |
| | SG-GSR | 80.98 | 77.03 | 89.07 | 75.29 | 88.08 | 71.90 | 62.80 | 84.06 | 68.14 | 76.30 | 58.37 | 48.70 | 55.85 | 33.40 | 77.38 |
| | ours | 84.61 | 82.93 | 82.53 | 82.98 | 86.63 | 72.50 | 65.59 | 84.26 | 73.13 | 71.75 | 80.49 | 79.65 | 90.40 | 88.38 | 74.48 |
| 10% | GCN_soft | 50.06 | 54.28 | 85.96 | 90.21 | 0.00 | 44.84 | 42.27 | 73.19 | 83.10 | 0.00 | 39.37 | 37.61 | 70.24 | 91.10 | 0.00 |
| | GCN_sig | 50.32 | 55.73 | 87.16 | 90.68 | 0.00 | 45.14 | 46.08 | 74.42 | 83.66 | 0.00 | 39.25 | 37.61 | 68.58 | 90.81 | 0.00 |
| | GCN_soft_$\tau$ | 78.14 | 73.52 | 85.95 | 72.73 | 84.88 | 67.71 | 59.82 | 78.05 | 59.28 | 77.60 | 64.84 | 64.03 | 67.83 | 64.04 | 65.44 |
| | GCN_sig_$\tau$ | 76.20 | 70.60 | 87.98 | 70.86 | 82.85 | 65.47 | 55.67 | 74.10 | 60.66 | 71.10 | 63.96 | 58.66 | 67.35 | 45.34 | 78.14 |
| | OpenWGL | 79.56 | 76.41 | 88.93 | 71.56 | 89.53 | 69.65 | 57.09 | 80.86 | 58.17 | 83.11 | 66.63 | 66.87 | 75.65 | 77.16 | 58.61 |
| | $\mathcal{G}^2Pxy$ | 78.78 | 76.34 | 76.95 | 79.72 | 77.61 | 71.59 | 61.81 | 70.58 | 72.02 | 71.10 | 57.55 | 40.34 | 54.56 | 19.21 | 86.74 |
| | STABLE | 69.08 | 60.00 | 83.10 | 67.60 | 70.93 | 65.32 | 56.06 | 76.19 | 68.42 | 61.69 | 69.49 | 66.48 | 74.20 | 59.91 | 76.77 |
| | RNCGLN | 68.18 | 57.96 | 79.05 | 71.33 | 64.24 | 68.76 | 64.11 | 78.65 | 80.06 | 55.52 | 63.58 | 62.27 | 67.77 | 63.59 | 63.56 |
| | SG-GSR | 76.07 | 71.46 | 84.91 | 70.16 | 83.43 | 72.94 | 64.61 | 83.55 | 74.52 | 71.10 | 63.74 | 59.21 | 67.60 | 49.34 | 74.70 |
| | ours | 81.24 | 75.64 | 89.40 | 76.46 | 87.21 | 72.35 | 63.95 | 81.92 | 68.42 | 76.95 | 76.52 | 75.64 | 88.23 | 85.24 | 69.88 |
| 20% | GCN_soft | 49.29 | 54.31 | 83.15 | 88.81 | 0.00 | 44.69 | 42.40 | 71.39 | 82.83 | 0.00 | 39.09 | 37.16 | 69.91 | 90.46 | 0.00 |
| | GCN_sig | 49.03 | 53.95 | 85.99 | 88.34 | 0.00 | 44.25 | 44.29 | 72.96 | 81.99 | 0.00 | 38.92 | 36.98 | 69.72 | 90.04 | 0.00 |
| | GCN_soft_$\tau$ | 75.81 | 66.31 | 83.69 | 63.40 | 91.28 | 67.12 | 57.54 | 72.76 | 55.96 | 80.19 | 67.01 | 62.87 | 70.67 | 52.00 | 78.43 |
| | GCN_sig_$\tau$ | 74.90 | 68.40 | 84.79 | 64.10 | 88.37 | 65.62 | 55.78 | 73.38 | 59.83 | 72.40 | 65.43 | 63.68 | 68.81 | 60.36 | 69.29 |
| | OpenWGL | 77.23 | 73.82 | 87.44 | 66.66 | 90.40 | 68.60 | 55.97 | 79.09 | 56.50 | 82.79 | 63.70 | 63.76 | 72.56 | 73.81 | 56.01 |
| | $\mathcal{G}^2Pxy$ | 77.36 | 74.71 | 74.12 | 78.08 | 76.45 | 70.15 | 57.99 | 68.09 | 67.42 | 67.59 | 57.23 | 35.43 | 53.50 | 11.91 | 91.73 |
| | STABLE | 61.84 | 43.12 | 76.70 | 55.71 | 69.48 | 65.72 | 55.78 | 79.78 | 68.98 | 62.34 | 70.87 | 68.01 | 75.43 | 61.61 | 77.92 |
| | RNCGLN | 68.05 | 57.72 | 78.65 | 71.33 | 63.95 | 66.97 | 63.30 | 76.46 | 78.67 | 53.25 | 62.77 | 60.69 | 67.01 | 60.42 | 64.56 |
| | SG-GSR | 79.56 | 74.62 | 88.77 | 71.10 | 90.12 | 71.15 | 60.56 | 81.71 | 65.65 | 77.60 | 61.22 | 56.23 | 63.97 | 44.99 | 73.58 |
| | ours | 79.95 | 73.58 | 87.53 | 75.29 | 85.76 | 72.12 | 64.80 | 84.79 | 72.30 | 72.73 | 76.18 | 75.21 | 87.52 | 85.08 | 69.04 |
| 30% | GCN_soft | 49.03 | 53.57 | 81.44 | 88.34 | 0.00 | 43.65 | 45.51 | 71.93 | 80.89 | 0.00 | 38.68 | 36.50 | 70.02 | 89.50 | 0.00 |
| | GCN_sig | 48.77 | 52.92 | 84.93 | 87.88 | 0.00 | 43.65 | 44.60 | 71.67 | 80.89 | 0.00 | 38.54 | 36.42 | 69.08 | 89.18 | 0.00 |
| | GCN_soft_$\tau$ | 74.64 | 67.86 | 82.58 | 63.87 | 88.08 | 65.17 | 56.72 | 73.37 | 57.34 | 74.35 | 64.89 | 62.74 | 68.60 | 58.44 | 69.80 |
| | GCN_sig_$\tau$ | 73.35 | 60.03 | 83.78 | 62.70 | 86.63 | 64.78 | 52.00 | 70.82 | 56.19 | 74.03 | 62.08 | 55.48 | 63.12 | 40.41 | 78.58 |
| | OpenWGL | 75.67 | 71.94 | 86.81 | 63.63 | 90.69 | 68.60 | 55.42 | 78.74 | 55.12 | 84.41 | 60.57 | 59.42 | 67.51 | 65.03 | 57.17 |
| | $\mathcal{G}^2Pxy$ | 75.67 | 72.76 | 72.55 | 76.22 | 75 | 69.05 | 58.80 | 67.02 | 67.31 | 71.10 | 56.87 | 30.58 | 52.59 | 06.14 | 95.49 |
| | STABLE | 60.03 | 43.27 | 83.00 | 58.74 | 61.63 | 61.88 | 42.77 | 71.32 | 50.97 | 74.68 | 67.58 | 63.08 | 68.60 | 51.97 | 79.45 |
| | RNCGLN | 67.27 | 56.91 | 78.50 | 69.93 | 63.95 | 62.63 | 60.01 | 73.90 | 78.12 | 44.48 | 63.09 | 62.37 | 67.43 | 66.95 | 60.15 |
| | SG-GSR | 80.47 | 75.97 | 88.53 | 72.73 | 90.12 | 71.00 | 63.41 | 82.50 | 72.02 | 69.81 | 63.27 | 58.48 | 66.35 | 48.58 | 74.46 |
| | ours | 79.17 | 75.05 | 87.04 | 74.83 | 84.59 | 71.98 | 63.62 | 81.02 | 68.98 | 76.62 | 74.72 | 73.98 | 84.53 | 84.15 | 67.54 |

Table 7: Comparison of Open-Set Node Classification: Test Accuracy (%), F1 Score (%), and AUROC (%) for Known Class Classification and Unknown Class Recognition on Three Citation Networks with One Unknown Class (u=1) Under random at Perturbation Rates ( 40%, 50%, 60% )

| Ptb Rate | Methods | Cora | | | | | Citeseer | | | | | Pubmed | | | | |
|---|---|---|---|---|---|---|---|---|---|---|---|---|---|---|---|---|
| | | tacc | F1 | auroc | kacc | uacc | tacc | F1 | auroc | kacc | uacc | tacc | F1 | auroc | kacc | uacc |
| 40% | GCN_soft | 48.25 | 51.65 | 82.55 | 86.95 | 0.00 | 42.90 | 44.06 | 73.75 | 79.50 | 0.00 | 38.47 | 36.22 | 68.72 | 89.02 | 0.00 |
| | GCN_sig | 48.38 | 52.01 | 84.28 | 87.18 | 0.00 | 43.05 | 40.40 | 67.04 | 79.78 | 0.00 | 38.32 | 36.13 | 68.73 | 88.66 | 0.00 |
| | GCN_soft_$\tau$ | 74.00 | 69.22 | 83.42 | 67.83 | 81.69 | 65.02 | 57.33 | 74.13 | 57.06 | 74.35 | 67.13 | 62.22 | 70.44 | 49.41 | 80.62 |
| | GCN_sig_$\tau$ | 72.96 | 63.25 | 83.99 | 65.50 | 82.27 | 65.02 | 53.90 | 72.90 | 53.46 | 78.57 | 65.58 | 62.01 | 68.03 | 53.22 | 74.99 |
| | OpenWGL | 75.54 | 69.52 | 86.06 | 63.86 | 90.11 | 68.01 | 54.72 | 78.42 | 54.84 | 83.44 | 61.28 | 61.26 | 69.80 | 70.64 | 54.15 |
| | $\mathcal{G}^2 Pxy$ | 75.03 | 70.21 | 70.41 | 73.42 | 77.03 | 67.49 | 58.63 | 67.64 | 69.42 | 66.66 | 56.78 | 24.14 | 51.21 | 0.0 | 1.0 |
| | STABLE | 61.84 | 45.10 | 78.72 | 56.88 | 68.02 | 61.43 | 52.78 | 68.47 | 60.66 | 62.34 | 67.10 | 62.71 | 68.86 | 52.39 | 78.31 |
| | RNCGLN | 67.40 | 57.00 | 78.12 | 70.40 | 63.66 | 62.18 | 59.50 | 73.59 | 77.29 | 44.48 | 61.28 | 58.79 | 66.03 | 58.76 | 63.20 |
| | SG-GSR | 78.91 | 75.21 | 87.66 | 76.22 | 82.27 | 71.90 | 64.59 | 82.03 | 73.96 | 69.48 | 64.27 | 60.38 | 66.60 | 52.03 | 73.58 |
| | ours | 77.36 | 72.58 | 86.13 | 71.79 | 84.30 | 71.75 | 63.09 | 80.91 | 64.82 | 82.14 | 74.97 | 74.12 | 86.59 | 84.98 | 67.34 |
| 50% | GCN_soft | 48.12 | 51.19 | 82.32 | 86.71 | 0.00 | 42.00 | 39.39 | 69.57 | 77.84 | 0.00 | 38.38 | 35.97 | 68.75 | 88.79 | 0.00 |
| | GCN_sig | 48.51 | 51.73 | 81.29 | 87.41 | 0.00 | 42.90 | 42.14 | 71.93 | 79.50 | 0.00 | 38.28 | 36.01 | 68.06 | 88.57 | 0.00 |
| | GCN_soft_$\tau$ | 72.45 | 67.81 | 81.96 | 63.64 | 83.43 | 65.02 | 53.57 | 72.32 | 53.19 | 78.90 | 65.14 | 61.41 | 66.19 | 52.03 | 75.12 |
| | GCN_sig_$\tau$ | 71.67 | 61.92 | 82.63 | 64.10 | 81.10 | 64.13 | 53.96 | 71.09 | 58.17 | 71.10 | 60.88 | 53.28 | 63.75 | 37.21 | 78.89 |
| | OpenWGL | 73.73 | 65.35 | 84.20 | 61.07 | 89.53 | 66.21 | 53.12 | 76.02 | 52.07 | 82.79 | 55.34 | 53.47 | 63.38 | 56.96 | 54.1 |
| | $\mathcal{G}^2 Pxy$ | 74.90 | 70.05 | 68.15 | 72.96 | 77.32 | 67.50 | 59.05 | 67.39 | 67.03 | 72.40 | 56.88 | 25.59 | 51.93 | 01.18 | 99.14 |
| | STABLE | 65.20 | 48.70 | 79.55 | 51.75 | 81.98 | 60.99 | 42.00 | 67.36 | 47.37 | 76.95 | 65.78 | 58.68 | 65.27 | 42.36 | 83.60 |
| | RNCGLN | 66.62 | 55.65 | 77.16 | 69.00 | 63.66 | 61.29 | 58.39 | 73.07 | 78.39 | 41.23 | 61.46 | 61.22 | 66.98 | 69.23 | 55.54 |
| | SG-GSR | 79.04 | 74.50 | 87.27 | 74.83 | 84.30 | 71.30 | 60.27 | 82.74 | 64.54 | 79.22 | 63.55 | 58.89 | 65.23 | 48.48 | 75.02 |
| | ours | 77.23 | 72.02 | 85.78 | 72.49 | 83.14 | 71.64 | 64.33 | 82.98 | 67.87 | 79.55 | 73.73 | 72.56 | 86.88 | 82.39 | 67.15 |
| 60% | GCN_soft | 47.99 | 51.63 | 82.32 | 86.48 | 0.00 | 42.75 | 42.63 | 73.07 | 79.22 | 0.00 | 38.03 | 35.63 | 69.48 | 87.99 | 0.00 |
| | GCN_sig | 48.38 | 51.80 | 81.10 | 87.18 | 0.00 | 42.45 | 41.17 | 71.94 | 78.67 | 0.00 | 37.85 | 35.64 | 67.25 | 87.58 | 0.00 |
| | GCN_soft_$\tau$ | 74.13 | 63.75 | 82.10 | 64.34 | 86.34 | 65.77 | 54.01 | 72.85 | 53.74 | 79.87 | 66.47 | 59.58 | 68.34 | 42.62 | 84.62 |
| | GCN_sig_$\tau$ | 72.45 | 61.18 | 81.81 | 63.40 | 83.72 | 65.17 | 53.67 | 72.17 | 54.85 | 77.27 | 64.67 | 59.95 | 66.66 | 48.29 | 77.14 |
| | OpenWGL | 72.83 | 62.96 | 82.86 | 58.27 | 90.98 | 65.77 | 52.61 | 76.77 | 51.80 | 82.14 | 59.23 | 59.44 | 68.80 | 69.64 | 51.30 |
| | $\mathcal{G}^2 Pxy$ | 75.03 | 63.34 | 65.76 | 67.59 | 84.30 | 67.12 | 57.39 | 66.64 | 67.03 | 68.24 | 56.78 | 24.14 | 5135 | 0.0 | 1.0 |
| | STABLE | 64.81 | 55.26 | 78.48 | 52.68 | 79.94 | 64.87 | 53.05 | 76.27 | 56.79 | 74.35 | 65.35 | 61.34 | 66.99 | 52.99 | 74.75 |
| | RNCGLN | 67.14 | 55.26 | 77.22 | 68.53 | 65.41 | 60.84 | 48.49 | 67.35 | 48.75 | 75.00 | 59.89 | 56.40 | 64.96 | 56.10 | 62.78 |
| | SG-GSR | 76.80 | 70.56 | 86.43 | 72.73 | 84.15 | 68.56 | 59.76 | 78.32 | 63.71 | 79.22 | 57.04 | 39.19 | 61.08 | 18.16 | 86.64 |
| | ours | 77.49 | 72.07 | 87.78 | 74.83 | 80.81 | 71.32 | 63.50 | 81.30 | 68.14 | 77.27 | 73.01 | 71.65 | 86.48 | 80.79 | 67.10 |

