# OpenReview forum: "OG-SNR : Open set graph learning with structural noise robustness"
_ICLR.cc/2026/Conference — Submitted to ICLR 2026_

### Official Review · Reviewer_DkBu · 2025-10-29

**Soundness:** 3
**Presentation:** 2
**Contribution:** 2
**Rating:** 4
**Confidence:** 4

**Summary:**

This paper introduces an innovative framework named OG-SNR, which, for the first time, systematically addresses the open-set node classification problem on graph-structured data, with a particular emphasis on robustness to structural noise. The proposed framework tackles this challenge through the integration of three key components. First, it optimizes the graph structure via node embedding refinement to enhance structural homophily. Second, a curriculum learning mechanism is incorporated to dynamically select reliable edges during training, thereby improving the model’s resistance to noise. Finally, by generating pseudo-unknown nodes and employing an entropy maximization loss, OG-SNR effectively distinguishes between known and unknown classes. Extensive experiments conducted on multiple benchmark datasets demonstrate that OG-SNR significantly outperforms existing baselines in both classification accuracy and open-set recognition capability, even under adversarial attacks and random structural perturbations.

**Strengths:**

1.	The new problem of "Open set Graph node Classification resistant to structural Noise" was proposed for the first time, combining the two key challenges of graph structural noise robustness and open set recognition, filling the gap in the field of graph learning.
2.	By organically integrating existing technologies such as graph structure learning, curricula-based edge learning, and entropy maximization, a two-stage framework (graph structure optimization + fine-tuning of curriculum learning) is proposed to form a unified framework, demonstrating the ability of creative integration.

**Weaknesses:**

1.	The experimental evaluation in this paper is entirely based on citation network datasets (Cora, Citeseer, PubMed), which exhibit high homogeneity in node types and relationship structures. To more comprehensively validate the generalization capability of the OG-SNR framework, it is recommended to extend experiments to more diverse graph structure scenarios, such as: social networks (e.g., Facebook, Twitter); information networks (e.g., protein interaction networks); e-commerce networks (e.g., Amazon co-purchasing networks).
2.	This paper effectively investigates the model's robustness to structural noise (edge-level perturbations), which is an important contribution. However, real-world graph data noise may also manifest in node features (e.g., missing features, outliers, or adversarial perturbations). The current experimental setup fails to evaluate the model's stability under such node-level noise. Therefore, it is recommended to supplement relevant experiments, such as adding Gaussian noise to node features, to verify whether the OG-SNR method possesses more comprehensive noise robustness.
3.	The proportion of unknown category nodes in the test set is a critical parameter. An excessively low proportion (e.g., 1%) may fail to adequately challenge the model's recognition capabilities, potentially leading to misleading high metrics. This paper neither controls nor specifies this proportion, making it difficult to accurately evaluate and reproduce the experimental results. We recommend conducting sensitivity analyses by systematically adjusting the proportion of unknown nodes in the test set and reporting the resulting trends in model performance. This would significantly enhance the rigor and persuasiveness of the experimental conclusions.
4.	The baseline methods selected in this paper encompass closed-set classification, open-set classification, and graph structure learning, ensuring comprehensive comparison. However, under the specific scenario of a “single unknown category,” the open-set node classification task is highly analogous to graph anomaly detection (where unknown nodes are treated as anomalies). Therefore, incorporating several classical or advanced graph anomaly detection methods as additional baselines enables a more comprehensive assessment of OG-SNR's performance level. This approach also highlights its advantages over general anomaly detection schemes when addressing specific open-set problems.
5. Please optimize the layout of Tables 4, 5, 6 and 7 to make them more conducive to reading.
6. It is suggested that the code be made public to facilitate the community's reproduction and comparison.

**Questions:**

Please refer to the weaknesses.

---

### Official Review · Reviewer_vzbw · 2025-10-30

**Soundness:** 2
**Presentation:** 3
**Contribution:** 2
**Rating:** 4
**Confidence:** 5

**Summary:**

This paper addresses the problem of open-set graph learning, aiming to classify known classes while identifying unseen ones under structural noise. The authors propose a framework that refines graph structures using both node features and topology, incorporates curriculum learning to mitigate noisy edges, and employs pseudo-OOD nodes with entropy-based optimization for open-set classification. The problem is clearly motivated and relevant to real-world scenarios involving uncertain structures. Experimental results are reported to support the framework’s effectiveness, but more diverse or challenging comparisons could strengthen the validation.

**Strengths:**

1. The problem studied in this paper is new.

2. The problem definition and technical descriptions are clear.

**Weaknesses:**

1. Some OOD samples may exhibit distributional shifts in their structural patterns. Would the proposed method mistakenly treat such OOD samples as structural noise?

2. The structure learning component appears overly simple, and the modules for structure learning and OOD detection are independent, making the framework seem like a combination of two separate problems. The paper provides limited analysis of whether the studied scenario presents unique challenges.

3. The datasets used are relatively simple. Larger and more comprehensive datasets, such as those from G-OSR: A Comprehensive Benchmark for Graph Open-Set Recognition, could be considered. In addition, the latest OSR baselines used for comparison are from 2023; newer methods should be included to better validate the effectiveness of the proposed approach.

4. The statement that “hard-to-learn edges are more likely to correspond to structural noise” lacks sufficient empirical or theoretical justification.

**Questions:**

Please see the weaknesses.

---

### Official Review · Reviewer_R2Nq · 2025-11-01

**Soundness:** 2
**Presentation:** 3
**Contribution:** 2
**Rating:** 2
**Confidence:** 3

**Summary:**

This paper studies the problem of open-set node classification on graphs under structural noise. The authors propose a new framework, OG-SNR, which refines graph structures using node similarity and applies curriculum learning to gradually include more complex edges during training. The method introduces an entropy maximization loss to improve the separation between known and unknown classes and automatically determines the confidence threshold for rejection. Experiments on standard benchmark datasets under both adversarial and random perturbations demonstrate consistent improvements over existing open-set and robust graph learning baselines.

**Strengths:**

The paper addresses an important and underexplored problem open-set graph learning with structural noise which is relevant to many real-world graph applications. The proposed method is clearly described, with a well-organized pipeline including structure refinement, curriculum-based edge learning, and entropy maximization. The experiments are conducted on standard datasets and show improvements over strong baselines in both accuracy and AUROC, supporting the claim that the approach enhances robustness and open-set detection.

**Weaknesses:**

The definitions of some key concepts are not clearly present, such as open set, OOD related concepts in the paper. The novelty of the proposed framework is limited, as it mainly integrates existing ideas such as structure refinement, curriculum learning, and entropy-based regularization into one system without a fundamentally new insight. The related work section does not comprehensively position OG-SNR against other recent open-set graph methods, particularly those that also consider noisy or uncertain structures. The experimental comparison lacks some important baselines e.g., robust GNNs, GNNs under distribution shifts. Moreover, the results are presented mainly as quantitative tables without strong qualitative or theoretical analysis to explain why the method works or which component contributes most beyond the ablation results, as far as I am concerned.

**Questions:**

How sensitive is the method to the choice of the pseudo-unknown node ratio and the noise parameters in entropy maximization?

Would the same framework generalize to heterogeneous graphs or large-scale real-world datasets beyond citation networks?

A more detailed analysis or visualization of learned structures could help clarify the effectiveness of the refinement process.

---

### Official Review · Reviewer_4FCi · 2025-11-04

**Soundness:** 2
**Presentation:** 3
**Contribution:** 2
**Rating:** 4
**Confidence:** 4

**Summary:**

This paper addresses open-set graph learning by proposing a framework that performs node classification while being robust to structural noise. The method refines graph structures by integrating node features and topology, employs curriculum learning to identify pseudo-OOD nodes, and uses entropy maximization for confidence-based open-set recognition. Experiments show that it is the first effective approach to handle open-set graph learning under structural noise.

**Strengths:**

1. The proposed open-set graph learning problem is interesting and meaningful.
2. The impact of structural noise in graphs is significant under the proposed setting.

**Weaknesses:**

1. The research problem is not clearly defined. The connection between structural noise and the proposed setting is unclear. For example, the paper mentions exploring “adversarial attacks and harsh changes such as out-of-distribution (OOD) nodes,” but it does not explicitly clarify how structural noise relates to these scenarios.

2. The proposed method lacks sufficient technical novelty. Combining node similarity to refine the graph structure and using curriculum-based learning to improve model robustness are already widely adopted techniques in graph learning. Moreover, the paper does not clearly justify the specific importance of these two components for handling structural noise.

3. The experimental evaluation is limited, as it only uses three basic datasets, which is insufficient to demonstrate generality.

4. The experimental setup lacks clarity. For instance, in Section 4.3, the paper varies the noise ratio but does not specify which type of noise ratio is being changed.

**Questions:**

on Weakness

---

### Meta-Review · Area_Chair_LXry · 2026-01-02

**Summary:**

This paper studies open-set node classification when graphs with noise. It includes a few major steps include structure refinement, edge learning, node sample, etc.

The reviewers agree the setting is interesting. The major concerns are more on the clear problem setting, which is connected to the definitipon of structural noise and noise ratio. Another common thought is that the solution is more integration. Finally, the results on just on three citation networks -- which is not sufficient clearly.

**Reviewer Concerns:**

As I mentioned above, the problems on problem setting, technical contribition, and insufficiency of datasets.

Since the authors did not do a rebuttal, nothing is addressed.

**Reviewer Scores:**

All will stay the same as there is no rebuttal.

---

### Decision · Program_Chairs · 2026-01-26

Reject